# Blockchain-Based Network Concept Model for Reliable and Accessible Fine Dust Management System at Construction Sites

Seungwon Cho [1], Muhammad Khan [1], Jaeho Pyeon [2] and Chansik Park [1,*]

1   School of Architecture & Building Science, Chung-Ang University, Seoul 06974, Korea;
    choseungwon@gmail.com (S.C.); muhammadkhan607@gmail.com (M.K.)
2   Department of Civil and Environmental Engineering, San Jose State University, San Jose, CA 95192, USA;
    jae.pyeon@sjsu.edu
*   Correspondence: cpark@cau.ac.kr; Tel.: +82-2-820-5263

**Abstract:** In total, 44.3% of particle matter 10 (PM10) is fugitive dust, and one of the main sources of fugitive dust generation in Korea is construction work (22%). Construction sites account for 84% of the total business places that have reported fugitive dust generation. Currently, the concentration of fine dust at construction sites is being remotely monitored by government inspection agencies through IoT sensors, but it is difficult to trust that appropriate fine dust reduction measures are being taken, because contractors can avoid taking these measures by submitting false reports or photos. In addition, since the fine dust monitoring system under government management is not an open platform and centralized system, residents near construction sites encounter difficulties in accessing information about fine dust. Therefore, in this study, we designed and constructed a blockchain network model to transparently and reliably provide network participants with the information associated with IoT data and fine dust reduction measures. To operate the blockchain network, we designed the chaincode, DApp, and network architecture. In addition, information on fine dust concentration and reduction measure photos were shared with the participants via the blockchain search tool (Hyperledger Explorer). The proposed blockchain network is expected to form a trust protocol among contractors, government inspection agencies, and citizens.

**Keywords:** dust management; construction; blockchain; decentralization; DApp; fine dust; IoT; distributed ledger technology

## 1. Introduction

In recent years, there has been a notable increase in social problems caused by fine dust generated at construction sites, and since 2013, when the World Health Organization (WHO) designated fine dust as a class 1 carcinogen, many countries have been attempting to reduce the amount of fine dust that they generate [1].

Fugitive dust accounts for 44.3% of all fine dust generated in Korea, and of this percentage, resuspended road dust, construction sites, and open areas generate fugitive dust rates of 45, 22, and 12%, respectively. Thus, the construction industry has the highest reported rate of fugitive dust generation. As of 2015, 84% of the business places reporting fugitive dust production were construction and civil projects [2]. Because the fine dust generated at construction sites is from temporary to demolition work, it is necessary to manage fine dust from the beginning to the end of construction projects [3,4].

In addition, there is an increase in the number of civil complaints regarding business places that generate fugitive dust, and most of these are related to the construction industry [5]. Construction sites are often located in places with a high population density, and the dust generated during construction works spreads to the surrounding environment [3,6]. Therefore, residents near construction sites suffer from the effects of fine dust. In 2016, 1697 out of 1750 fine dust-generating workplaces in Seoul were in construction sites, and 3748 out of 3771 complaints regarded fine dust. In addition, civil complaints associated with

construction dust have been continuously increasing, and they exceeded 20,000 in 2015 [7]. This increasing trend indicates that the current strategies for fine dust management at construction sites are ineffective.

The current method of managing fine dust at construction sites in Korea is to install internet of things (IoT) sensors at large-scale construction sites (total floor area of 100,000 m$^2$ or a business area equal to or exceeding 90,000 m$^2$) using administrative district and monitoring dust emissions in real time [8]. The amount of fine dust generated at construction sites is measured quarterly or monthly. However, remote real-time monitoring has helped to solve the scarcity of government inspection agencies measuring fine dust at construction sites [9]. This method can thoroughly assess the amount of and changes in the generated fine dust during construction, contributing to achieving the goal of an immediate reduction in fine dust [10,11]. However, based on a centralized server, the current fine dust management system may result in information imbalance [12]. As an example, From January 2015 to June 2017, the amount of fine dust generated by China's five major cities (Beijing, Shanghai, Shenyang, Guangzhou, and Chengdu) was significantly lower than the value measured by the US embassy [13]. This proved that it is difficult to trust the information provided by the Chinese local government, which centrally manages information. In addition, Chinese citizens who are aware of this fact also point out that they do not trust information reported by local governments.

Recently, the construction industry has been establishing various policies to reduce the generation of fine dust [14]. However, various organizations are participating, and due to the "information imbalance" caused by the vertical contract structure, it is difficult to implement effective policies based on reliable information [15]. The centralized computer system aggravates it, and it mainly causes distrust among government inspection agencies, contractors, and citizens.

Although IoT measurement systems enable government inspection agencies to remotely monitor construction sites at the district or city level, contractors ultimately implement fine dust reduction; thus, appropriate implementation of fine dust reduction measures is under question [16]. For instance, a contractor can avoid taking fine dust reduction measures by submitting false photos or reports [17].

Moreover, accessing information related to fine dust monitoring systems is difficult for the public. Since the current monitoring system is a centralized system, there is no responsibility to share information on fine dust sensors at construction sites, and manipulation of the information provided to the public may occur [13,18]. Therefore, it is difficult to remotely read the fine dust measurements at construction sites and determine whether the contractor carried out the fine dust reduction measures.

The problems of real-time monitoring of fine dust in current construction sites through IoT can be summarized into two main categories. First, due to the centralized server, fine dust concentration information is hidden, and the information provided by the system can be tampered with. Second, a specific method to verify whether fine dust reduction measures are implemented is needed.

Therefore, this study presents a blockchain-based network concept model for the development of a reliable and accessible fine dust management system at construction sites. Because blockchain can fundamentally prevent data forgery and share information through a distributed ledger technology and decentralization, it is a suitable solution for the current monitoring system. The proposed method stores fine dust concentration data and pictures of fine dust reduction measures in the blockchain. Users can freely participate in the fine dust management network and transparently view the workflow information. Blockchain and decentralization are expected to solve the information imbalance problem and form a trust protocol.

## 2. Literature Review

First, the literature review focuses on how the IoT sensor is used for fine dust management at construction sites. Then, the technology that can be utilized as an alternative is defined via analysis of the limitations of the current management method. This is followed by an examination of the applicability of the technology. Since the current monitoring system is based on IoT, we determined accessibility and reliability as the weaknesses of the IoT system. Thus, as the solution to centralization, blockchain can be used on the basis of many use cases. Table 1 is a comparative table that summarizes the purpose of previous research and the differences with the current study to gain a better understanding.

### 2.1. IoT Sensors for Fine Dust Management at Construction Sites

As measuring fine dust and preparing reports are performed by humans, data falsification and manipulation are possible. In addition, the time consumption and human resources required to visit construction sites for fine dust measurements and writing and submitting a report are considerable. Therefore, many studies using the IoT to minimize human intervention, remotely measure the concentration of fine dust at construction sites in real time, and establish advanced monitoring systems have been conducted.

Smaoui et al. suggested real-time air dust monitoring that can be deployed ubiquitously at a construction site and integrated as part of a daily construction management system [19].

A. Carbonari et al. developed a wireless pervasive and real-time dust concentration monitoring system to maintain admissible concentration values defined by national legislation in place [20].

M. Khan et al. employed two low-cost dust sensors (Sharp GP2Y1014AU0F and Alpha sense OPC-N2) without implementing control measures to explicitly evaluate, compare, and gauge them for construction activities. Latin hypercube sampling was used to analyze the measurement results and predict the exposure concentrations [21].

Real-time monitoring of fine dust at construction sites using IoT has been studied from various perspectives. Many studies focus on the accuracy of the fine dust measurement of IoT sensors for construction sites. However, few studies exist on the accessibility and reliability of information collected through these fine dust-monitoring systems.

### 2.2. Progress in Alleviating Limitations of the Centralized Network

The existing construction dust monitoring system for government inspection agencies is managed using a central database system [22]. It is difficult to access the centralized system to read important information without permission from the system manager; all stakeholders are not fairly treated from the point of view of transparency [23].

According to the study conducted by Reyna et al., security problems such as data falsification exist, as the IoT data are recorded in a central network. Moreover, free browsing of the recorded IoT information on a single server is difficult. Therefore, we introduced a new distributed IoT network to solve these problems. One approach to increase trust in accessing IoT data is through distributed services trusted by all the participants, because it guarantees the immutability of the data [24].

Wang et al. explored the application of blockchain to the IoT in Industry 4.0 using its security tools and technology. The employed sensor network has a client/server structure in which all data are managed using a central server. This structure has security and privacy problems because it is always connected to the network. To solve these problems, the importance of decentralization must be emphasized. Blockchain was suggested as an appropriate technology for the development of the IoT network [25].

Conoscenti et al. explained that information asymmetry between information providers, owners, and viewers occurs because centralizing organizations can monopolize the information. Their study suggested the application of a blockchain information management structure. The basic idea is to safely store the data produced by personal IoT devices

on a distributed system whose design authorizes the public (the real data owners) to decide what to share and with whom [26].

In centralized networks, sharing information is systematically difficult because it is concentrated on a single server. This structure causes information asymmetry, and a trust protocol cannot be primarily formed between network participants [27]. To solve the problems of such a centralized network, some studies have emphasized the importance of decentralization. In addition, many studies are being conducted to secure the reliability and accessibility of the network using the blockchain as a representative technology of decentralization.

**Table 1.** Publications on the application of IoT and blockchain.

| Authors | Application Area | Security | Reliability | Trust | Decentralization | Automation | Transparency | Scalability | Integrated Digital Technologies | Study Type | Industry |
|---|---|---|---|---|---|---|---|---|---|---|---|
| [18] | Dust monitoring and visualization in BIM | | √ | | | √ | √ | | IoT, BIM | Experiment | Construction |
| [19] | Real-time monitoring system of dust concentration | | | | | √ | | √ | IoT | Experiment | Construction |
| [20] | Real-time monitoring system of dust concentration | | | | | √ | | √ | IoT | Experiment | Construction |
| [24] | Blockchain for industrial IoT | √ | | √ | √ | | √ | √ | IoT, blockchain | Theory | Construction |
| [25] | Privacy and decentralization in the IoT | √ | | √ | √ | | | √ | IoT, blockchain | Theory | All Industries |
| [28] | supply chain in ready mixed concrete | | √ | | √ | √ | √ | | RFID, blockchain | Theoretical model proposal | Construction |
| [29] | Payment automation for project stakeholders | | √ | √ | | √ | √ | √ | Blockchain | Theoretical model proposal | Construction |
| [30] | BIM data modification provenance | | √ | √ | √ | √ | √ | | BIM, blockchain | Theoretical model proposal | Construction |
| [31] | Quality information management | | √ | | √ | √ | √ | √ | BIM, sensor module, blockchain | Theoretical model proposal | Construction |
| [32] | Lightweight smart dust IoT security system | √ | | √ | | | | √ | IoT, blockchain | Experiment | All Industries |
| **This research** | **Fine dust management system with blockchain** | √ | √ | √ | √ | √ | √ | √ | **IoT, blockchain** | **Theoretical and practical** | **Construction** |

### 2.3. Securing Reliability and Share of Information Using Blockchain Technology

Blockchain, as a decentralizing technology, can enhance the reliability and efficiency of information storage and management by setting up a trust protocol that can implement an open data sharing network [28]. In addition to cryptocurrency, blockchain technology is used in various applications. In the construction industry, it is mainly studied to improve transparency, scalability, and security.

Lanko et al. employed blockchain technology using RFID tags. It can be observed when creating a single database where suppliers and customers of ready-mixed concrete can receive actual and reliable data on turnover, placing offers, finding customers, optimizing the production, and using the ready-mixed concrete [29].

Luo et al., proposed a methodology to automate construction payments by formalizing smart contracts and applying blockchain technology. This framework also addresses the conditions required for security in construction projects, such as confidentiality and information integrity, in multi-party environments [30].

Zheng et al. proposed a novel BIM system model called bc-BIM to address information security in mobile cloud architectures. In particular, bc-BIM is proposed to facilitate BIM data audits for historical modifications using blockchain in the mobile cloud with big data sharing. It guarantees tracing and authentication and prevents interference with historical

BIM data. Simultaneously, it can generate a unified format to support future open sharing, data audits, and data derivation [31].

Zhong et al. proposed a blockchain-based framework for construction quality information management, extending the blockchain applications in construction quality management. They confirmed that blockchain could facilitate trust in construction quality management by providing distributed, encrypted, and secure information records and supporting automated compliance checking of construction quality [32].

J. Park et al. proposed a lightweight blockchain scheme that helps device authentication and data security in a smart dust monitoring BIM environment. This study suggested that blockchain can significantly reduce the time required for smart dust device authentication [33].

The construction industry utilizes the features of blockchain-distributed ledgers, smart contracts, asymmetric encryption, and network access rights to make stored information unmodifiable and transparently disclosable, in addition to providing network participant verification functions. Therefore, blockchain can solve information asymmetry among the contractor, sub-contractor, government, and public. In addition, blockchain can be used as a basic technology to improve the existing fine dust monitoring system at construction sites.

However, the literature on blockchain networks for fine dust sensors in construction sites is limited. Moreover, disclosing the records of fine dust reduction actions performed by contractors to governmental inspection agencies or the public in a decentralized environment has not been studied. In this paper, we present a blockchain-based fine dust network model for construction sites. Additionally, we aimed to design and implement a network model to transparently and reliably provide the key information associated with the IoT data and fine dust reduction to network stakeholders for remote monitoring.

## 3. Research Objectives

Blockchain can fundamentally prevent data manipulation. A decentralized system can share information with network members. This chapter explains how blockchain technology works to make a reliable and accessible fine dust management network.

### 3.1. Securing Reliable Information of Fine Dust at Construction Sites Using Blockchain Technology

To improve reliability in fine dust management at construction sites caused by the manipulation of measured values at construction sites, blockchain technology is required to record fine dust management information in a decentralized environment; thus, it cannot be arbitrarily modified and can be shared to all participants in the blockchain network. The main data to be recorded in the blockchain are "fine dust sensor concentration information", "picture of fine dust concentration reduction action", and "fine dust concentration regulation at the construction site". The regulation is not data generated from the construction site, but for the smooth operation of the network, it is stored in the blockchain to determine whether the excessive fine dust is generated or not. By reviewing information stored in the blockchain, government inspection agencies and local citizens can trust that the measurements of fine dust concentration at the construction site and fine dust reduction actions have been appropriately performed, and remote inspection can be carried out to solve the issue of the shortage of inspectors.

### 3.2. Improvement in Accessibility to Fine Dust Management Information at Construction Sites

As the current fine dust management system for construction sites is centralized for government inspection agencies, residents near construction sites are not aware of the amount of fine dust generated at the construction sites unless the government inspection agencies provide information. Decentralization using a distributed ledger of blockchain technology allows the public to view the data stored in the distributed ledger and share the information recorded in the blockchain with participants. This can form a trusted network consisting of the contractors who reduce the amount of fine dust at construction sites, inspection agencies who monitor the concentration of fine dust, and civic groups who are curious about fine dust information from construction sites near their homes.

## 4. Blockchain-Based Fine Dust Management System Model for Construction Sites

In Figure 1, The proposed workflow is similar to the methodology currently used by local governments to manage construction sites in terms of output; however, a "process black box" functions as a decentralized network by applying blockchain technology. This process regards trusting distributed and stored information until it is created and viewed through the participant network configuration based on a trust protocol.

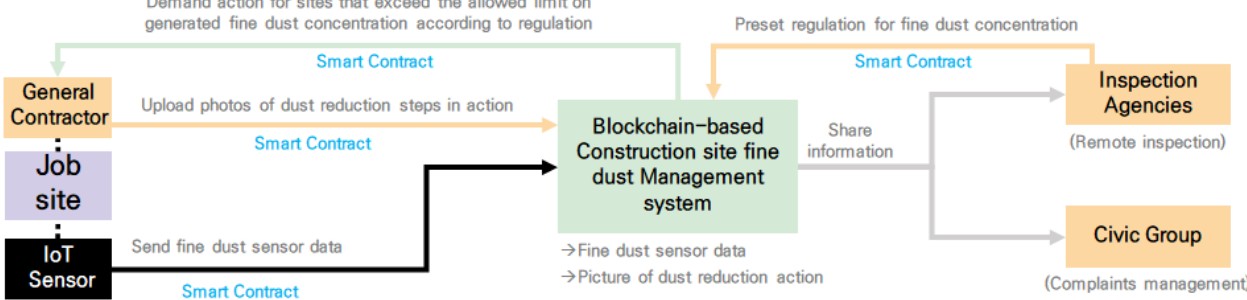

**Figure 1.** Workflow of the proposed fine dust management system.

Blockchain is a distributed storage technology that makes data falsification almost impossible; however, trust will be compromised if forgery occurs when data enter the blockchain. To improve the reliability of the stored IoT data and to solve the oracle problem [34], the proposed system semi-automates a series of processes until the IoT data collected at construction sites are automatically stored in the blockchain to minimize human error. The digitalized fine dust concentration measurement values are automatically verified as reliable data on the system (smart contract) without third-party verification [35]. An inspection agency sets the fine dust concentration regulation value for construction sites in advance. Moreover, an inspection agency can manage construction sites flexibly by considering the surrounding environment or the laws imposed by local governments and set a certain value for each construction site or jurisdiction. Therefore, the concentration of fine dust at a construction site is evaluated based on this regulation value and IoT data. When the determination result is "exceeded," the system automatically (smart contract) notifies the network participants to reduce fine dust concentrations. Subsequently, the contractor uploads photos of the fine dust reduction steps in action, including stopping scattering dust generation, water sprinkling, installation of dust barriers, and dust filtering, which are used as evidence of fine dust reduction by the contractor. The above series of fine dust management at construction sites can improve the efficiency of inspection agencies using automation and increase the reliability of the information. In addition, residents can be transparently aware of applied fine dust management regulations at nearby construction sites and fine dust concentration reductions, making it easier to process public complaints associated with the fine dust. Consequently, a contractor can reduce wasting time and manage human resources to perform additional fine dust management-related tasks.

### 4.1. System Architecture

Figure 2 is the proposed system, a blockchain network that uses the "Hyperledger Fabric platform". It is possible to create and manage each blockchain by allocating one construction site to one blockchain of the channel through the multi-channel function. In addition, by utilizing the MSP (membership service provider) and PKI (public key infrastructure), network participants can use application functions.

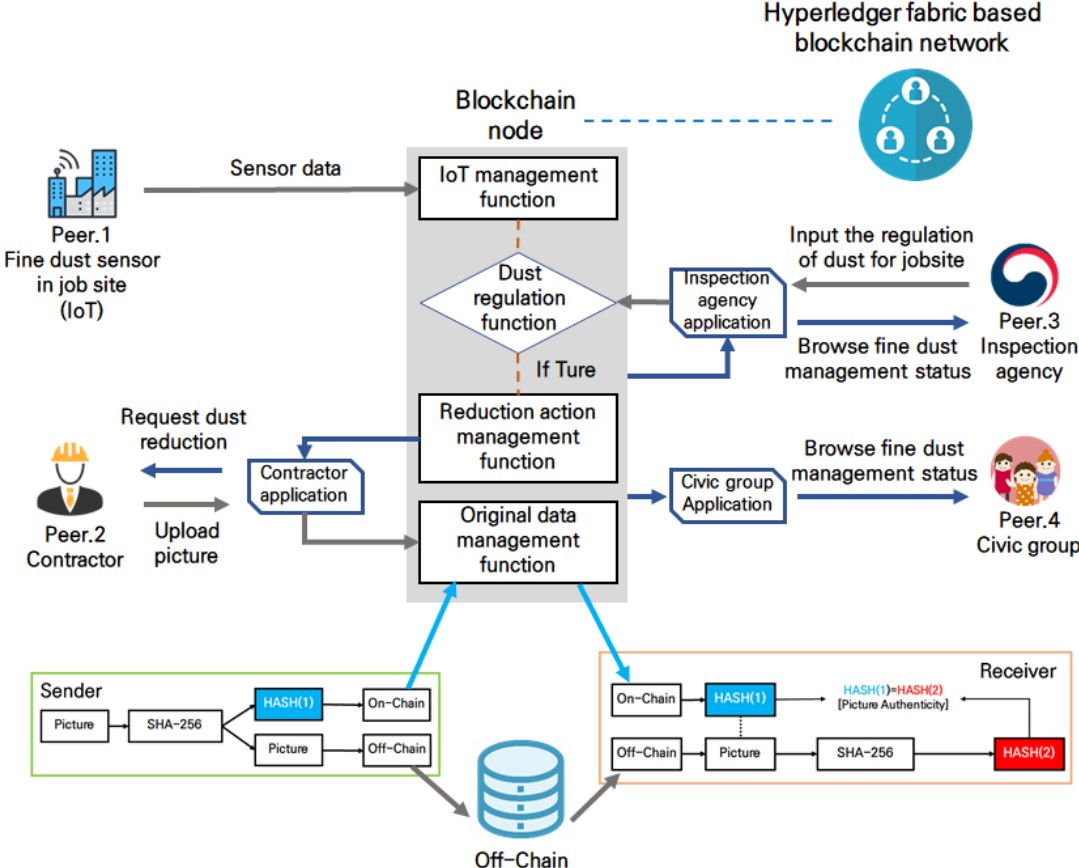

**Figure 2.** The system architecture of the proposed fine dust management system.

The JSON format value generated using the IoT installed at the construction site is transmitted through wireless transmission protocols, such as Wi-Fi and Bluetooth, and it is stored in a distributed ledger. Because the photos uploaded by the contractor are large-capacity information, converting the original photos into digital data and using them on the blockchain is challenging. The HMac (hash-based message authentication code algorithm) [36] mechanism was applied to solve this problem. The basic operation principle is similar to that of the HMac algorithm; however, the information sent and received is a picture. Hash values, which are low-capacity data corresponding to photos, are extracted using the SHA-256 (Mac function) and uploaded to the blockchain. The recipient can check the photo's integrity by regenerating the original photo's hash value and comparing it with the hash value of the photo uploaded to the blockchain in order to determine the similarity between the two hash values. As a contractor may upload pre-taken or inappropriate photos, the photos are taken and uploaded simultaneously in connection with the camera API.

Decentralized applications (DApps) running on the blockchain are divided into contractors, inspection agencies, and civic groups in terms of their roles. A contractor receives a notification of corrective action and uploads a photo using a DApp. Inspection agencies and civic groups can view fine dust management at construction sites through the DApp. In addition, inspection agencies have the authority to manage the setting of fine dust concentration regulations for construction sites.

### 4.2. Chaincode Design

Chaincodes were designed to reduce oracle problems caused by human intervention and to automate systems. A chaincode consists of four functions that store and manage necessary data, excluding the functions essential for system operation. The proposed chaincode (smart contract) and the data to be managed on the blockchain are defined as illustrated in Table 2.

**Table 2.** Chaincode classification.

| Chaincode ID | Invoke Authority | Description | Key Value |
|---|---|---|---|
| IoT management function | IoT | Transmits fine dust concentration values at construction site | {'IoTID', 'Value', IoTtime'} |
| Original data management function | Contractor | Extracts and saves hash value of uploaded photo | {'GCID', 'Gctime', 'HashValue', 'ImageUrl'} |
| Dust regulation function | Government Inspection Agency | Inspection agency inputs the fine dust concentration regulation value in advance | {'RegulationID', 'RegulationValue'} |
| Reduction action management function | System | Rings alarm to inform contractor to reduce fine dust when fine dust concentration exceeds the regulation value | {'GCID', 'Value', RegulationValue'} |

The subjects that utilize the chaincode are IoT, contractors, government inspection agencies, and systems. The methodology of using the chaincode according to each subject is discussed below.

The IoT for measuring the concentration of fine dust generated at construction sites uses the chaincode of the system (smart contract) as an "input argument." The main IoT data to be stored in the blockchain are IoT ID, Value, and IoTtime. The unique ID of the IoT (IoT ID) is used to identify in which construction site the fine dust concentration was measured to explain the key value of the sensor data. To determine the hourly fine dust concentration, IoTtime is recorded.

For the contractor to upload a photo of the fine dust reduction measures using the application, the hash value of the original photo data is automatically extracted, and GC ID, Gctime, Hash Value, and Image Url are saved. The GC ID is used to determine which contractor uploaded the photo and whether the contractor was participating in the blockchain network. Because image data consume a large amount of capacity, the hash value corresponding to the photo is extracted using the SHA-256 algorithm. The original picture that was stored in the off-chain is linked by Image Url for open access.

An inspection agency inputs the [regulation value] of fine dust concentration using the application. [Regulation ID] is used to prove that the inspection body is a reliable participant.

If the fine dust concentration value at a construction site exceeds the set value, the system automatically alerts the construction company to take required fine dust reduction actions. GC ID is used to specify the contractor and Value and Regulation Value are used to prove that the set value has been exceeded.

*4.3. Dapp Design*

A DApp refers to a client application that can employ a chaincode from outside the network and utilize its functions. Figure 3 displays the process of using chaincode through the DApp and the code information of the function that the client finally uses on the UI. After the chaincode is installed by a peer and initiated, its functions can be utilized with a DApp. To create or identify data on the blockchain, "init" is first run to check the existence of chaincode, and the chaincode ID is determined using the "invoke" function. Furthermore, IoT management, dust regulation, and original data management functions are executed when a client makes a request through a proposal. The reduction management function is conditionally executed on the system according to the IoT and regulation values. It is also possible to provide inspection agencies and civic groups with real-time information regarding fine dust management at construction sites using the query function, a basic function of the chaincode's application. To prove that a client is a verified user (Peer) of the network, the Client ID (GC ID, Regulation ID, and Civicgroup ID) is examined using the SDK, and when using a DApp, it is checked before submitting the proposal using the MSP.

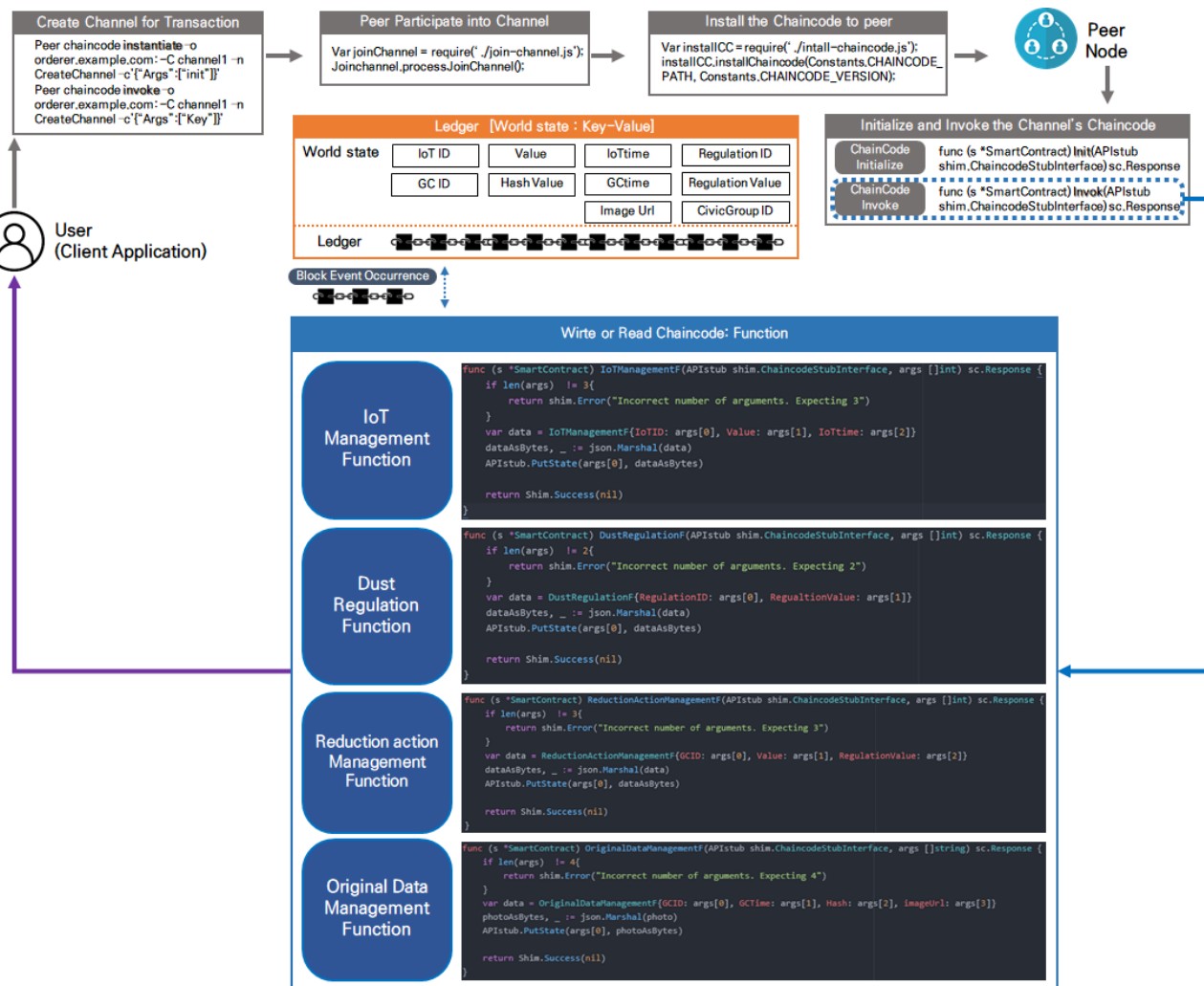

**Figure 3.** System chaincode and function code.

Figure 4 displays the user interface (UI) and the code of the decentralized application used to communicate with the blockchain network. Figure 4a shows the screen that an inspection agency uses to set the fine dust concentration regulations at construction sites. First, a client's participation in the network is verified using the wallet, and the construction site (Channel) is selected to input the value. Concentration regulations can be set for each construction site or jurisdiction. Figure 4b shows the screen of the notification being sent to the contractor when the fine dust concentration at the construction site exceeds the set value. Before uploading photos, the client's identity (ID) is checked using the wallet.

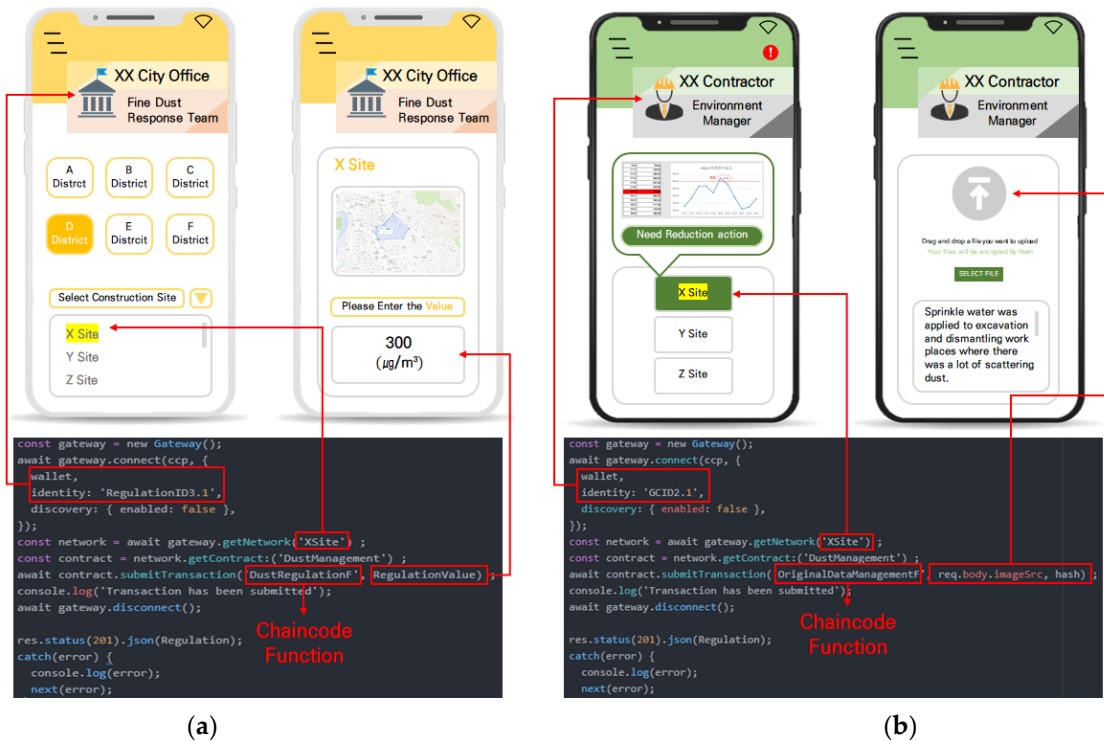

**Figure 4.** (**a**) DApp for government inspection agency; (**b**) DApp for contractor.

### 4.4. Network Design

On-site fine dust information is stored and shared on each blockchain, in which four Org (IoT, contractor, inspection agency, and civic group) peers participate. Figure 5 shows how peers participate in blockchain (Channel) and authenticate each other's identity. The channel MSP function provided by the "Hyperledger Fabric" forms a network between the participant Org groups. For peer identity authentication, a pair of public and private keys and enrollment certificates that prove participation in the corresponding channel are generated based on the public key infrastructure (PKI). In addition, the verifiable ID is registered as a member of the blockchain network of the corresponding channel using the channel MSP. The orderer, who creates blocks by organizing transactions, is selected by the Org of the government inspection agency with access to all channels. To select the leader peer, who delivers the latest blocks created to the network, the spinning method, which randomly selects a leader among the Org peers, is used.

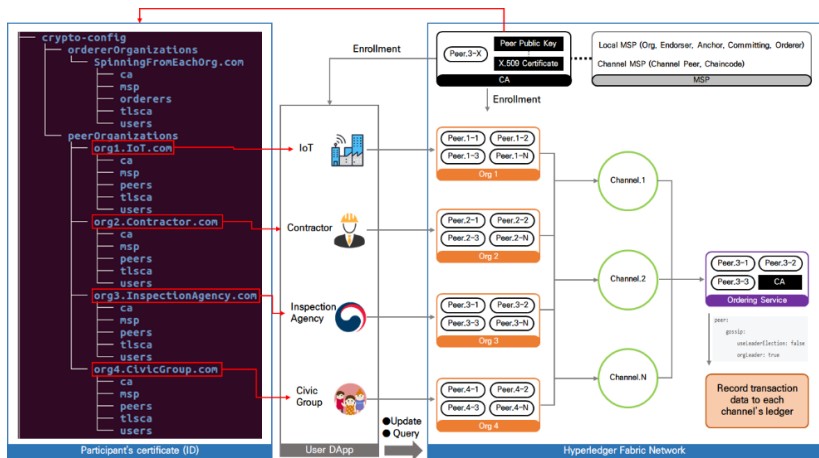

**Figure 5.** The network structure for peer and channel.

## 5. Implementation

The environment of the system proposed in this study is based on the Ubuntu operating system and runs on Hyperledger Fabric version 1.4. HTML, CSS, JavaScript, and Node.js are used as development languages for network implementation. World State DB is used to exchange application and blockchain data, and Next.js is used for SPA.

A virtual scenario was simulated to operate the proposed blockchain system with theoretical background knowledge for fine dust management in actual construction sites. The detailed assumptions for running the simulated scenario are as follows:

1.  The peers currently participating in the X construction site (channel 1) are the IoT Node (IoTID:1-1), contractor (GCID:2-1), inspection agency (RegulationID:3-1), and civic group (Civicgroup ID: 4-1). These peers are configured as virtual nodes.
2.  Methods for acquiring fine dust concentration data at construction sites include dynamic measurement using robots, drones, etc., and static measurement via the installation of sensors in each section. In this scenario, the static measurement using a light scattering digital dust sensor was considered for real-time measurement of the fine dust concentration in a construction site. It was assumed that the sensor performed the measurement in intervals of 10 min, and the GP2Y1010AU0F module was used to acquire the sensor data [21].
3.  Fine dust regulations vary by country or region. This scenario was simulated by referring to the "Special Act on Fine Dust Reduction and Management of Korea." Inspection agencies set the fine dust concentration value to 300 $\mu g/m^3$.

### 5.1. Operation

The basic workflow and the transaction process scenario considering assumptions are shown in Figure 6a, which is a diagram expressing the workflow for the management of fine dust at a construction site using this system, and it has a simple and linear structure. However, because IoT sensor data are periodically generated, "End" means that the construction has been completed. Figure 6b shows the flow of data and transaction flow in each work process, and it is a diagram reconfigured according to the assumption based on Figure 2.

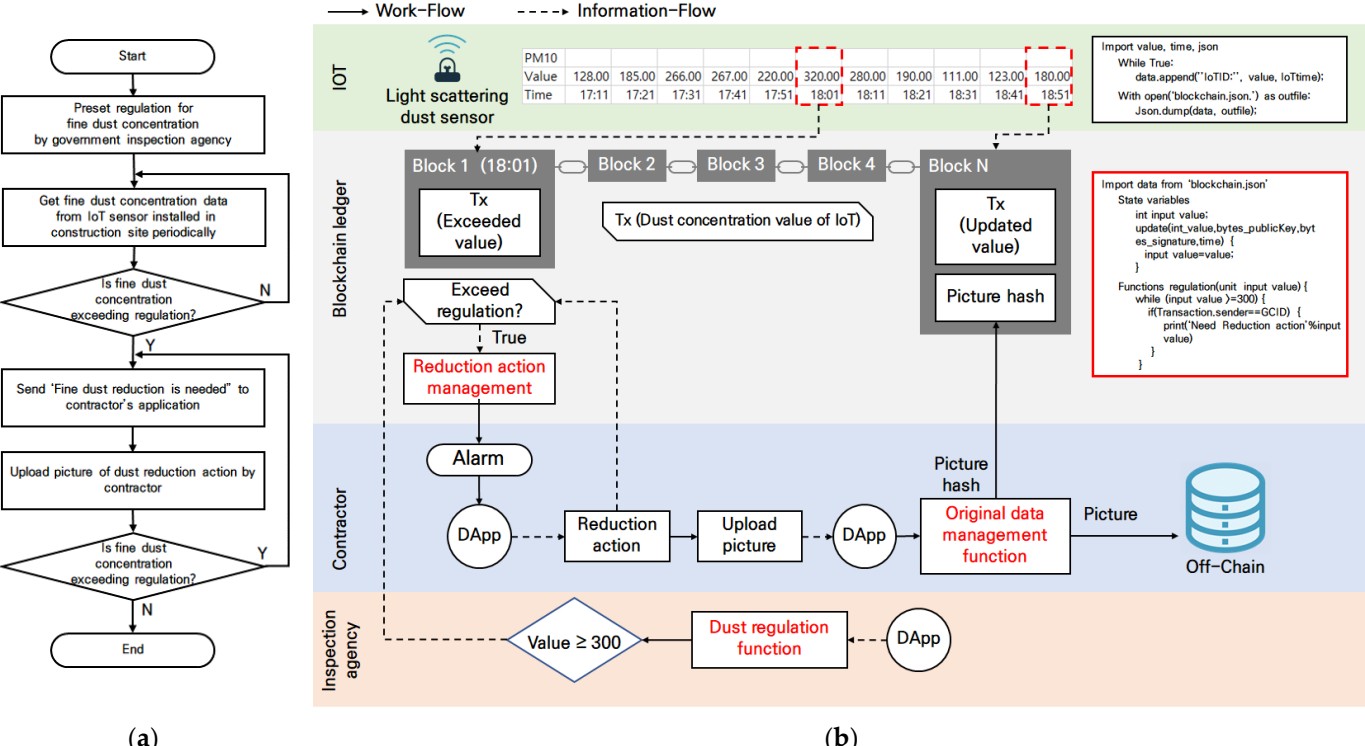

(**a**)    (**b**)

**Figure 6.** (**a**) Workflow; (**b**) transaction scenario.

First, the government inspection agency (RegulationID:3-1) sets the fine dust concentration regulation at the construction site as 300 μg/m$^3$ through the DApp. Then, the IoT sensor (IoTID:1-1) transmits the fine dust concentration value. Since the fine dust concentration value is 320 μg/m$^3$ at 18:01 and exceeded 300 μg/m$^3$, it automatically alarms the construction company (GCID:2-1) through the Reduction Action Management Function. The contractor implements fine dust reduction measures, and when a picture is uploaded, the hash value of the picture is stored in the blockchain. In this process, the information generated by the IoT, contractor, inspection agency, and system is recorded on the blockchain and can be viewed through various tools.

### 5.2. Results

As blockchain data are difficult for users to view, an explorer tool is needed to query block information, transaction information, network node information, chaincode, and information stored in the ledger. Hyperledger Fabric utilizes the PostgreSQL database to provide users with the blockchain information as visualized data using the explorer. Therefore, the Hyperledger Explorer was used to read recorded data in the blockchain instead of code. Figure 7 shows the block information of the result data presented in Figure 6.

Figure 7 provides the basic blockchain network-related information and transaction information. Peer information provide the list of network participants. One peer of Org, which corresponds to the IoT, contractor, inspection agency, and civic group, participates in the blockchain network. In addition, the MSP is activated to confirm the access authority of participants to the channel Xsite. Transaction information provides fine dust management information uploaded by each participant to the channel's blockchain. It shows that Xsite information was generated by the IoT, inspection agency, contractor, and system. Transaction information includes details, and the transaction detail (key value) from each workflow is shown in Figure 8.

In the transaction details, input data are recorded in a key-value format. The key is a fixed value as specified in Section 4.2, and the value changes as the participants input it. According to Figure 8, the government inspection agency inputted 300 μg/m$^3$ as a regulation value. When the IoT sensor received a value exceeding 300 μg/m$^3$ on 2 July 2021/18:01:04, the system automatically sent an alarm message to the contractor. The alarm content contained information on who received the data and why the alarm message was sent. When the contractor uploads picture for dust reduction, the hash value and image Url of the picture are recorded.

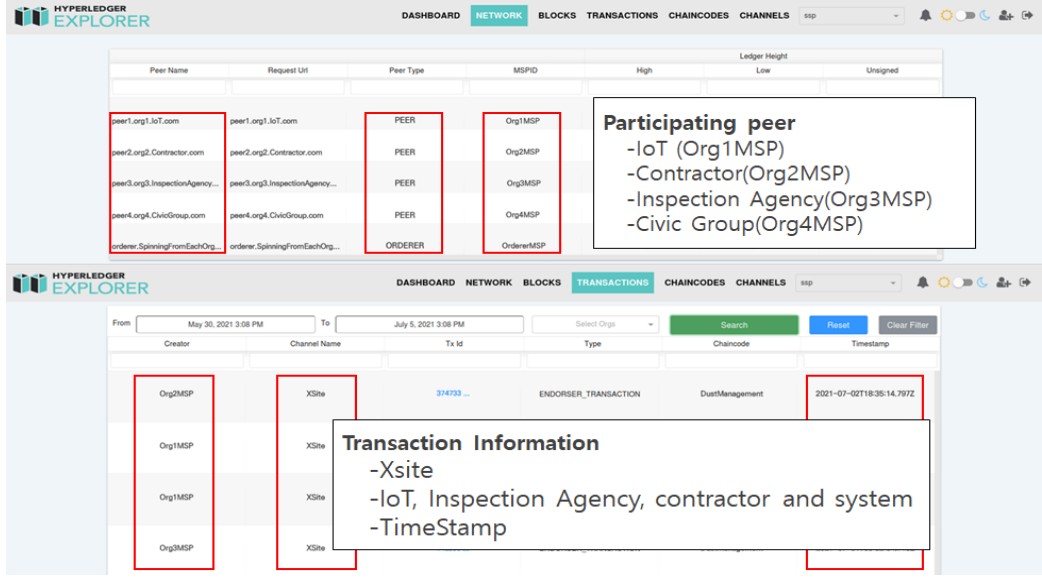

**Figure 7.** Peer and transaction information.

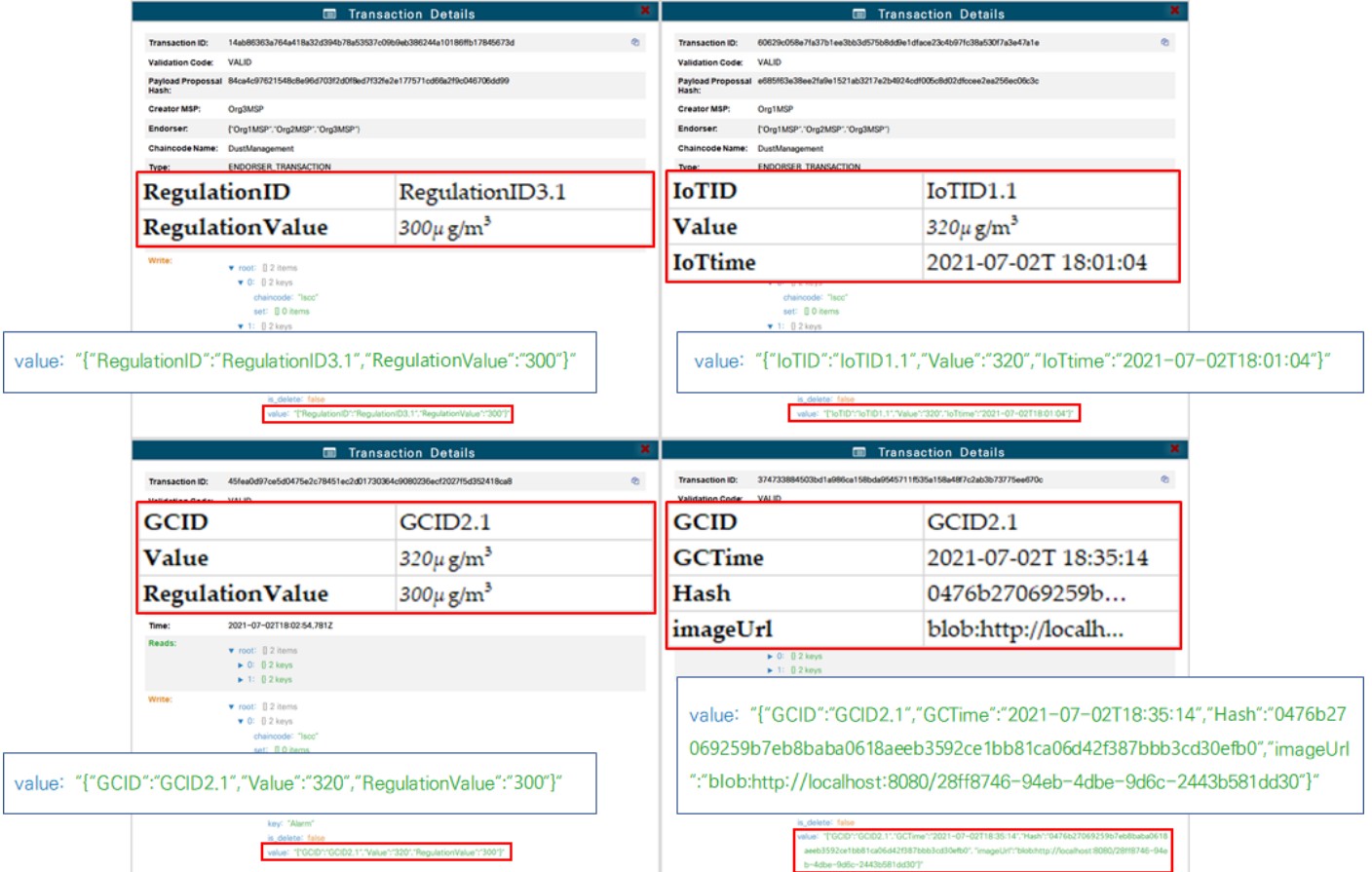

**Figure 8.** Transaction details for key-value data.

Fine dust at construction sites must be measured in real time through IoT for immediate reduction, and information must be shared in real time for remote monitoring. Although the current system utilizes IoT, it is not an open system. There are many manual tasks, such as sending a message to reduce fine dust or on-site inspection by the government agency. However, this study automated a significant task, and all information recorded in the blockchain network is disclosed to participants. Table 3 shows a comparison between the existing system and the proposed system. The results are the average values of 15 experiments. As the aim of this study is reliability, rather than improving the accuracy of the IoT sensor for the measurement of fine dust concentration at the construction site, it verifies the speed of the work process that can be implemented in real time for data sharing. Similar to the current system, the fine dust IoT measurement system of the local government of Seoul, South Korea, is an object of comparison. According to Table 3, the proposed system reduces the time required for work through automation, and the required time is reasonable. In addition, it enhances reliability through the photo upload function and photo forgery detection. As shown in Figure 8, the data can be shared through transaction details in real time.

Table 4 shows the qualitative comparison evaluation results of the proposed system and the existing system. In terms of reliability, the proposed system provides decentralization based on blockchain. blockchain Information cannot be falsified through HMac or a distributed ledger, so network participants can trust and share information. In addition, human intervention was minimized through chaincode. In terms of efficiency, the work process was automated. In addition, new construction sites and participants can easily participate in the network by using the provided chaincode, multi-channel, and MSP.

**Table 3.** Comparison of the work processes.

| Classification | Current System | Suggestion System |
|---|---|---|
| Input fine dust concentration regulation | manually | manually |
| IoT data transmission | less than 1 s | less than 1 s |
| Time to determine excess fine dust concentration | less than 1 s | 0.7 s |
| Time to send dust reduction alarm | manually | 5.3 s (automatically) |
| Inspection time | visit construction site | 9.6 s (upload picture) |
| Forgery check | none | 7.2 s (HMac, automatically) |
| Monitoring service | only government inspection agency | all participants |

**Table 4.** Qualitative evaluation.

| | Evaluation | Current System | Suggestion System |
|---|---|---|---|
| Reliability and Accessibility | Decentralization | Low | High |
| | Security | Normal | High |
| | Reliability | Normal | High |
| | Trust | Normal | High |
| | Transparency | Low | High |
| Efficiency | Automation | Normal | High |
| | Scalability | Normal | High |

## 6. Discussion

The proposed system can reduce the moral hazard while updating information, because the blockchain "process black box" works in all processes. Moreover, participants who read the recorded data can solve the information imbalance problem using a distributed ledger to minimize the reverifying task of validating the information's reliability. As a result of this process and output, participants who read data, such as inspection agencies and civic groups, can use reliable data to determine the fine dust concentration, breach concentration limits, and implement measures in order to reduce the concentration of fine dust. Therefore, it is possible to form a trust protocol among the contractor, who reduces fine dust at construction sites; the inspection agency, who monitors the concentration of fine dust; and the public, who are curious about information regarding fine dust being produced at construction sites near their homes. A private blockchain platform was established to classify the authority to access the blockchain network based on the participants. The platform utilizes "Hyperledger Fabric," which has a multi-channel function. It can manage multiple distributed ledgers; thus, it can easily manage various construction sites. For the use of the multi-channel, the inspection agency participates in the channel of a city or town, the contractor participates in the channel of construction sites that are self-managed, and the civic group selectively participates in the channel of construction sites near the residential area. Therefore, it is possible to easily create and view information by participating as a peer in the blockchain network.

Consequently, residents near construction sites are guaranteed access to fine dust management information. In addition, civil complaints can be reduced by using transparent and reliable information. Moreover, inspection agencies will implement policies to improve the quality of public life by calculating dust emission charges and deploying fine dust management teams. Additionally, contractors can gain a reputation and enhance their management skills by improving their autonomous and transparent environmental management capabilities. This can be an advantage when applying the preliminary qualification system for bid qualifications and compensation to extend construction periods [37].

### 7. Conclusions

This paper proposed a blockchain-based network concept model for the development of a reliable and accessible fine dust management system at construction sites. The proposed system compares the inspection agency's fine dust concentration regulation value with its corresponding value at real construction sites transmitted using the IoT to evaluate fine dust generation. Subsequently, the responsible contractor is automatically notified about the fine dust reduction steps that need to be taken. Then, the contractor captures and uploads photos associated with the implementation of the fine dust reduction steps.

The main contribution of this study is the design of a chaincode to implement the proposed system and the design of a DApp, a client application running on the blockchain network. In addition, a network design for system modularization was presented so that contractors, inspection agencies, and civic groups could conveniently participate in the network. To verify the effectiveness of the proposed system operation, we confirmed that the blocks were created, and information was stored in the Hyperledger Explorer based on a hypothetical scenario.

In the future, we intend to expand the system function to manage fine dust concentrations at large-, medium-, and small-sized construction sites. Subsequently, the proposed system will support other sensors so that fine dust concentration sensors and various types of construction site management sensor platforms can be applied based on blockchain technology. The system will be provided as a protocol-based tool that can be implemented on the web and smartphones. The following considerations are required to develop the expanded system in the future:

1. Various sensors that can measure fine dust at construction sites and the corresponding error rates should be considered. For instance, light-scattering digital dust sensors are widely used because of their low price. However, because it is sensitive to temperature, humidity, and wind, the error range is extensive; thus, it is used only for indoor applications, and the accuracy of the measurement results is under question.
2. Regulations such as measurement standards for fine dust concentration at construction sites, sensors, and punishments are not clearly defined. Therefore, it is difficult for inspection agencies to set regulation values for the fine dust concentration at construction sites.
3. At construction sites, various types of tasks generate fine dust depending on the type of construction. Therefore, the location and amount of fine dust differ depending on the scale and type of construction. Accordingly, it is necessary to optimally locate the fine dust measurement sensors and optimize the autonomous mobile measurement method, such as the robot dog and drone.
4. At construction sites, various factors, such as fine dust, temperature, humidity, and noise, are managed in real time via IoT sensors. In the future, based on the methodology of this study, the proposed blockchain system will be deployed to separate more parameter information from dust, such as noise, accidents, and safety issues. Therefore, it is necessary to study the use cases of various IoT sensors and establish methodologies for each case.

**Author Contributions:** Conceptualization, methodology, and original draft, S.C.; data curation and interpretation, M.K.; review and editing, J.P.; project administration and funding, C.P. All authors have read and agreed to the published version of the manuscript.

**Funding:** This study was financially supported by the National Research Foundation of Korea (NRF) grant funded by the Korean Government Ministry of Science and ICT (MSIP) (No.NRF-2019R1A2B5B02070721) and supported by the Chung-Ang University Research Grants in 2020.

**Institutional Review Board Statement:** Not applicable.

**Informed Consent Statement:** Not applicable.

**Data Availability Statement:** Not applicable.

**Conflicts of Interest:** The authors declare no conflict of interest.

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
