# Peer review of "Blockchain-Based Network Concept Model for Reliable and Accessible Fine Dust Management System at Construction Sites"

_applsci, doi:10.3390/app11188686_

Round 1
Reviewer 1 Report
The manuscript is about dust control in the working places and very interesting for everyone who works in the offices, especially those who work near the building place.
The main shortcoming of the manuscript is:
The text has not convinced me that mast use blockchain to achieve the goal. It looks like that they are using the truck to carry some small things. Maybe the subject is too narrow, and authors should consider the proposed system as a universal system for managing many (maybe all) properties (not only dust) of the environment.
Other shortcomings:
There are many mistakes in spelling, language, and use the plural. There are some typos. I recommend checking text by the native-speaking man.
References are not in the expected format.
References of figures 2,3 and 4 in the text are after figures.
Subchapter titles are direct after chapter titles. For instance, chapter 2 and subchapter 2.1, 3 and 3.1 and rest. Readers would like a short introduction to the chapter.
What is the purpose of the division into very short subsections (e.g., 4.3.2, 4,3.3, and others)? That suggests that are much different thought threads.
Reviewer 2 Report
This study suggests a blockchain-based network concept model for fine dust management system at construction sites.
The manuscript is well written and falls with the scope of the journal.
The introduced contribution is the design of a chaincode to implement the proposed system and the design of DApp.
The related works are recent and well explained. Future directions of the study are well detailed in the conclusion.
Above some recommendations to enhance the quality of the manuscript:
- The investigation of the related works can be achieved using comparative table(s) to better understand the differences/drawbacks/advantages of the previous studies.
- The study mainly focus on the design of a model (for fine dust management) without detailing the experimental process and its results. Numerical tests should be achieved to validate the proposed modeling.
Round 2
Reviewer 1 Report
Now the manuscript is good but there are still having spelling problems, for instance:
row 87: hided -> hidden (perfect tense),
row 150: viewers occurs -> viewers occur (plural without "s" ),
and other,
but now it is much better.
